# Epsilon-Near-Zero Absorber by Tamm Plasmon Polariton

**Rashid G. Bikbaev** [1,2,*] **, Stepan Ya. Vetrov** [1,2] **and Ivan V. Timofeev** [1,2]

[1] Kirensky Institute of Physics, Federal Research Center KSC SB RAS, Krasnoyarsk 660036, Russia; s.vetrov@inbox.ru (S.Y.V.); ivan-v-timofeev@ya.ru (I.V.T.)

[2] Siberian Federal University, Krasnoyarsk 660041, Russia

\* Correspondence: bikbaev@iph.krasn.ru

**Abstract:** Two schemes of excitation of a Tamm plasmon polariton localized at the interface between a photonic crystal and a nanocomposite with near-zero effective permittivity have been investigated in the framework of the temporal coupled-mode theory. The parameters of the structure have been determined, which correspond to the critical coupling of the incident field with a Tamm plasmon polariton and, consequently, ensure the total absorption of the incident radiation by the structure. It has been established that the spectral width of the absorption line depends on the scheme of Tamm plasmon polariton excitation and the parameters of a nanocomposite film. The features of field localization at the Tamm plasmon polariton frequency for different excitation schemes have been examined. It has been demonstrated that such media can be used as narrowband absorbers based on Tamm plasmon polaritons localized at the interface between a photonic crystal and a nanocomposite with near-zero effective permittivity.

**Keywords:** photonic crystal; nanocomposite; epsilon near-zero material; Tamm plasmon polariton

## 1. Introduction

Tamm plasmon polaritons (TPPs) are a special type of electromagnetic surface states, in which the field decays exponentially on each side of the surface [1] and the energy transfer along the surface can be stopped. This state can be experimentally observed as a narrow [2,3] or broadband [4] resonance in the optical transmission or reflection spectrum of a sample at wavelengths inside the band gap of a photonic crystal (PhC). Among the proposed and implemented TPP applications are organic solar cells [5], absorbers [6], lasers [7], sensors [8], integrated optical networks [9], heat emitters [10], and spontaneous emission amplifiers [11]. The high degree of field localization at the TPP frequency makes it possible to enhance second-harmonic generation [12] and implement the extremely high light transmittance through a subwavelength hole [13]. The TPPs and devices based on them are designed, as a rule, using a planar metallic film coupled with a PhC. The potentialities of optimizing the optical properties of such structures by means of the variation in the film parameters are exhausted by choosing the material and thickness of this film. New opportunities are offered by metasurfaces and metal-dielectric nanocomposites (NCs), i.e., artificial media structured in a special way, used as film materials [14]. A nanocomposite is a dielectric matrix with metallic particles uniformly distributed over its volume, which is characterized by the resonant effective permittivity. The optical properties of initial materials have no resonant features [15–17]. The position of a resonance in the visible spectral range, as well as the frequency band where the NC behaves like a metal are determined by the effective permittivity. The latter, in turn, depends on the permittivities of initial materials and the concentration, shape, orientation, and size of metallic particles. In particular, the authors of [18] first demonstrated

the possibility of forming a localized mode at the interface between a PhC and an NC representing a transparent matrix with uniformly-dispersed silver nanoparticles.

It is worth noting that metal-dielectric NCs can serve as materials with the near-zero effective permittivity. It should be noted that the properties of ENZmaterials are determined by their dispersion. For example, the real part of the dielectric permittivity may be less than the imaginary one by two or more orders of magnitude. Moreover, the ENZ material can be received in the visible, UV, and IR spectral ranges [19–21]. The classical plasmonic materials (silver, gold) do not have such dispersion properties. For this reason, ENZ materials are interesting for various plasmonics applications, for example for tunable broadband absorbers [22].

In recent years, such materials have been in the focus of researchers [23] due to the possibility of controlling the wave front shape [24], amplifying the light transmission through a subwavelength aperture [25], and enhancing the nonlinear effects [26,27]. Moreover, tunable and broadband absorbers based on ENZ materials have been realized in [22,28]. Furthermore, these materials can be used for TPP formation [29]. However, to the best of our knowledge, the possibility of realizing ENZ absorbers based on TPP has not been considered to date. This structure compared to metasurface absorbers [30,31] has the advantage of thickness, which allows for efficient removal of heat and protects the device from damage. Furthermore, TPP can be coupled with other types of localized modes [32–34], which allows controlling its spectral properties, and can be used for the creation of tunable absorbers.

Therefore, in this work, we show the possibility of designing narrowband absorbers based on the TPPs localized at the interface between a PhC and an NC with the near-zero permittivity. Using the temporal coupled-mode theory, we compare two TPP excitation schemes and predict the spectral properties of the localized modes. We demonstrate that the analytical results agree well with the numerical calculation.

## 2. Model Description

We consider a PhC structure, which represents a layered medium bound by a finite NC layer (Figure 1). The PhC unit cell is formed from materials $a$ and $b$ with respective layer thicknesses $d_a$ and $d_b$ and permittivities $\varepsilon_a$ and $\varepsilon_b$. The nanocomposite layer with thickness $d_{eff}$ and permittivity $\varepsilon_{eff}$ consists of metal nanospheres uniformly distributed in a transparent matrix made of optical glass.

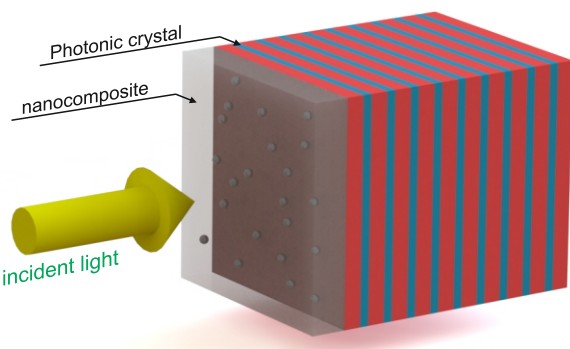

**Figure 1.** Schematic of a one-dimensional photonic crystal (PhC) coupled with a nanocomposite (NC) layer with the near-zero permittivity.

Then, we assume the PhC to be placed in a vacuum.

Maxwell's equations are reduced to the Helmholtz equation. For a *p*-polarized wave propagating in the *xz* plane of a layered medium of isotropic materials stacked along the *z* direction, we have:

$$\left[\frac{d^2}{dz^2} + \left(\frac{\varepsilon\omega^2}{c^2} - k_x^2\right)\right] E_x(z) = 0,$$
$$H_y(z) = \frac{-i\omega}{c}\frac{\varepsilon dE_x/dz}{c(\varepsilon\omega^2/c^2 - k_x^2)}, \quad (1)$$

where $k_x = n(\omega/c)$ is the tangential wavenumber along the *x* axis, *c* is the speed of light, and $\omega$ is the frequency. The effective permittivity of the NC can be described by the Maxwell–Garnett formula [35]:

$$\varepsilon_{eff} = \varepsilon_d \left[1 + \frac{f\left(\varepsilon_m(\omega) - \varepsilon_d\right)}{\varepsilon_d + (1 - f)\left(\varepsilon_m(\omega) - \varepsilon_d\right)1/3}\right], \quad (2)$$

where *f* is the filling factor, i.e., the volume fraction of nanoparticles in the matrix; $\varepsilon_d$ and $\varepsilon_m(\omega)$ are the permittivities of the matrix and nanoparticle metal, respectively; and $\omega$ is the radiation frequency. We determine the permittivity of the nanoparticle metal using the Drude approximation:

$$\varepsilon_m(\omega) = \varepsilon_0 - \frac{\omega_p^2}{\omega^2 + i\omega\gamma}, \quad (3)$$

with the parameters of silver being $\varepsilon_0 = 5$, $\omega_p = 9$ eV, and $\gamma = 0.02$ eV [36,37]; for glass, we have $\varepsilon_d = 2.56$.

For certainty, we assume alternating PhC layer materials to be silicon dioxide ($SiO_2$) with a permittivity of $\varepsilon_a = 2.10$ and zirconium dioxide ($ZrO_2$) with a permittivity of $\varepsilon_b = 4.16$. The respective layer thicknesses are $d_a = 74$ nm and $d_b = 50$ nm.

As an example, Figure 2 shows the *Re* $\varepsilon_{eff}$ and *Im* $\varepsilon_{eff}$ dependences calculated using Formula (2) with regard to Equation (3) at $f = 0.11$.

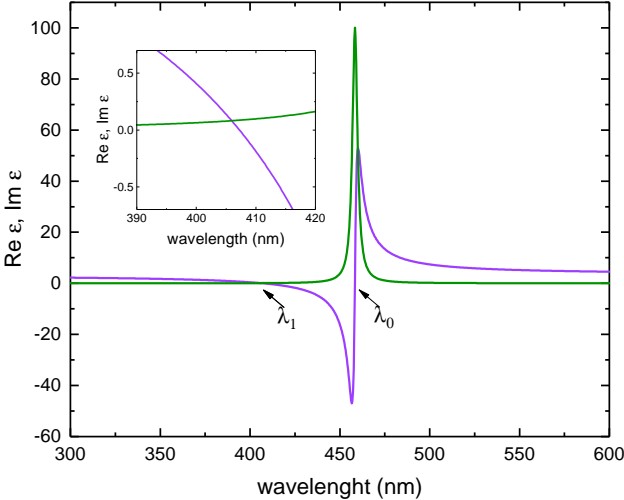

**Figure 2.** Dependences of the real *Re* $\varepsilon_{eff}$ (purple line) and imaginary *Im* $\varepsilon_{eff}$ (green line) parts of the effective permittivity $\varepsilon_{eff}$ on the incident light wavelength. The filling factor is $f = 0.11$.

It can be seen that in the range of $[\lambda_1, \lambda_0]$, we have *Re* $\varepsilon_{eff} < 0$; i.e., the NC behaves like a metal. At wavelengths of $\lambda_0 = 458.4$ nm and $\lambda_1 = 407.2$ nm, the real part of the effective permittivity takes the zero value. In this case, $\lambda_0 = 458.4$ nm is the resonant wavelength, at which the *Im* $\varepsilon_{eff}$ value is

maximum; therefore, the localized modes cannot form in the vicinity of this point. In view of the above, in this study, we investigate the TPPs in the vicinity of the point $\lambda_1$.

## 3. Temporal Coupled-Mode Theory

The optical properties of the TPP can be described using the temporal coupled-mode theory [38–41]. This theory is grounded on the fact that any mode (resonance) can be characterized by eigenfrequency $\omega_0$ and number $N$ of the ports through which the energy passes into this mode and leaks from it. The energy loss in the channels is described by the relaxation times $\tau_l$, $l = 1...N$, and the mode is described by the complex amplitude $A$ related to the amplitudes $s_{l\pm}$ of the incoming and outgoing energy flows. These quantities are described by the ordinary differential equation:

$$\frac{dA}{dt} = -i\omega_0 A - \sum_{l=1}^{2} \frac{A}{\tau_l} + \sum_{l=1}^{2} \sqrt{\frac{2}{\tau_l}} s_{l+}, \tag{4}$$

The relation between the flow amplitudes is:

$$s_{l-} = -s_{l+} + \sqrt{\frac{2}{\tau_l}} A. \tag{5}$$

If the resonance is excited through one port with $s_{l+} = -s_{l+}e^{-i\omega t}$, the resonance amplitude can be expressed as:

$$A_l(\omega) = \frac{\sqrt{\frac{2}{\tau_l}}}{i(\omega - \omega_0) + \sum_{l=1}^{N} \frac{1}{\tau_l}} s_{l+}. \tag{6}$$

The amplitudes of reflection from port $l$ to port $l'$ form the scattering matrix:

$$r_{ll'} = \frac{s_{l'-}}{s_{l+}} = -\widehat{\delta}_{ll'} + \frac{\sqrt{\frac{2}{\tau_l}}\sqrt{\frac{2}{\tau_l'}}}{i(\omega - \omega_0) + \sum_{l''=1}^{N} \frac{1}{\tau_l''}}, \tag{7}$$

where $\widehat{\delta}_{ll'}$ is the Kronecker symbol.

These reflectances and transmittances correspond to the spectral peaks in the form of Lorentz contours with the full width at half maximum:

$$2\gamma = \sum_{l=1}^{N} \frac{1}{\tau_l}. \tag{8}$$

For one channel, we have $l = l'$. In this case, the reflection amplitude can be written as:

$$r_l = -1 + \frac{2\gamma_l}{i(\omega_0 - \omega) + \sum \gamma_{l'}}. \tag{9}$$

The zero reflection is only possible under the critical coupling conditions, i.e., at $\omega = \omega_0$:

$$r_l(\omega = \omega_0) = 0. \tag{10}$$

This is sufficient to describe the spectral properties of the TPPs using the temporal coupled-mode theory.

## 4. Study of the Spectral Properties of the TPP Using the Temporal Coupled-Mode Theory

The TPP formation is contributed to by three energy channels, each characterized by the amplitude relaxation rate $\gamma$, which is the ratio between the power of energy relaxation to a channel and the energy accumulated in the TPP. We denote the rates of energy relaxation to the NC transmission and absorption channels and PhC transmission channel as $\gamma_{NC}$, $\gamma_A$, and $\gamma_{PhC}$, respectively. Since the energy accumulated in the TPP is the same for determining the rate of relaxation to each channel, the relaxation rates and corresponding energy coefficients of the structure are expressed by the proportion:

$$\gamma_{NC} : \gamma_A : \gamma_{PhC} = T_{NC} : A_{NC} : T_{PhC}. \tag{11}$$

We study the conditions of the critical coupling between the TPP and incident field for two excitation schemes.

If one of the mirrors is opaque, we may ignore one of the energy relaxation channels. Then, critical coupling condition (10) for the scheme of excitation through the NC layer (Figure 3a) can be written in the form:

$$\gamma_{NC} = \gamma_A; \gamma_{PhC} = 0 \Leftrightarrow T_{NC} = A_{NC}; T_{PhC} = 0, \tag{12}$$

and, for the scheme of excitation through the PhC (Figure 3b), in the form:

$$\gamma_{PhC} = \gamma_A; \gamma_{NC} = 0 \Leftrightarrow T_{PhC} = A_{NC}; T_{NC} = 0. \tag{13}$$

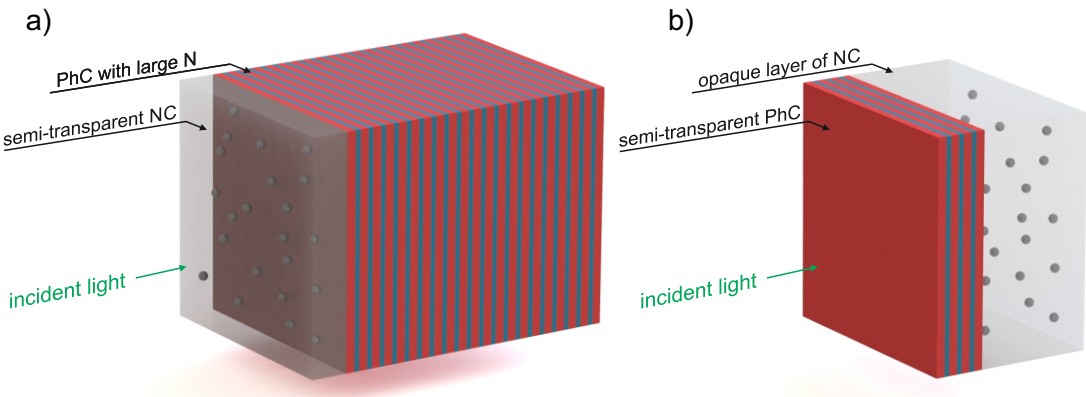

**Figure 3.** Schematic of the Tamm plasmon polariton (TPP) excitation from the side of (**a**) an NC film and (**b**) a PhC.

Critical coupling Equation (12) can be solved graphically. To do that, we should plot together the dependences of the transmittances and absorptances of the NC film on its thickness and incident radiation wavelength. The intersection of the surfaces corresponds to the equality of the coefficients and critical coupling. The energy coefficients for the NC layer can, in turn, be determined using the Airy formula derived for the metallic layer [42]. For this purpose, we consider a plane-parallel NC film with the refractive index $n_2 = n_2' + in_2''$ placed between two dielectric media with the refractive indices $n_1 = n_{SiO_2}$ and $n_3 = 1$. The transmittances, reflectances, and absorptances of the NC film will be determined as:

$$T_{NC} = \frac{n_3}{n_1} \left| \frac{t_{12} + t_{23}e^{i\beta}}{1 + r_{12}r_{23}e^{2i\beta}} \right|^2, \quad R_{NC} = \left| \frac{r_{12} + r_{23}e^{2i\beta}}{1 + r_{12}r_{23}e^{2i\beta}} \right|^2,$$
$$A_{NC} = 1 - T_{NC} - R_{NC}; \tag{14}$$

where $\beta = 2\pi n_2 d_{eff}/\lambda_0$ is the phase incoming when the wave passes through the layer, $\lambda_0$ is the wavelength, $d_{eff}$ is the layer thickness, and $t_{12} = 2n_1/(n_1 + n_2)$, $r_{12} = (n_1 - n_2)/(n_1 + n_2)$, and $t_{23} =$

$2n_2/(n_2+n_3)$, $r_{23} = (n_2-n_3)/(n_2+n_3)$ are the transmission and reflection amplitudes at the 1-2 and 2-3 interfaces, respectively.

The NC film spectra calculated using Formula (14) are shown in Figure 4.

The critical coupling is presented by the line of intersection of these surfaces. According to the numerical calculation, Condition (12) in the wavelength ranges of ($407 < \lambda < 408$ nm), where the NC effective permittivity is near-zero, is satisfied at NC film thicknesses of about 200 nm. Thus, coupling of a PhC with the NC film of such a thickness will lead to the formation of the TPP, at the wavelength of which the entire radiation incident on the structure will be absorbed.

The graphic solution of the critical coupling conditions for two TPP excitation schemes is illustrated in Figure 5.

In this case, the dependence of the PhC transmittance on the number of periods $N$ was determined as:

$$T_{PhC} = 1 - |r_N|^2, \tag{15}$$

where $r_N$ is the amplitude of reflection from the multilayer structure [43]:

$$r_N = \frac{CU_{N-1}}{AU_{N-1} - U_{N-2}}, \tag{16}$$

$U_N = \sin\left[(N+1)K\Lambda\right]/\sin[K\Lambda]$ and $K = 1/\Lambda \arccos\left[(A+D)/2\right]$ is the Bloch wavenumber, and $N$ is the number of periods. $A$, $B$, $C$, and $D$ are the elements of the $2 \times 2$ transfer matrix, which relates the plane wave amplitudes in Layer 1 of the unit cell to the analogous amplitudes for an equivalent layer in the next PhC unit cell.

$$
\begin{aligned}
A &= e^{ik_{1z}d_a}\left[\cos k_{2z}d_b + \tfrac{1}{2}i\left(\tfrac{k_{2z}}{k_{1z}} + \tfrac{k_{1z}}{k_{2z}}\right)\sin k_{2z}d_b\right], \\
B &= e^{-ik_{1z}d_a}\left[\tfrac{1}{2}i\left(\tfrac{k_{2z}}{k_{1z}} - \tfrac{k_{1z}}{k_{2z}}\right)\sin k_{2z}d_b\right], \\
C &= e^{ik_{1z}d_a}\left[-\tfrac{1}{2}i\left(\tfrac{k_{2z}}{k_{1z}} - \tfrac{k_{1z}}{k_{2z}}\right)\sin k_{2z}d_b\right], \\
D &= e^{-ik_{1z}d_a}\left[\cos k_{2z}d_b - \tfrac{1}{2}i\left(\tfrac{k_{2z}}{k_{1z}} + \tfrac{k_{1z}}{k_{2z}}\right)\sin k_{2z}d_b\right],
\end{aligned}
\tag{17}
$$

where $k_{1z} = (\omega/c)\sqrt{\varepsilon_a}$ and $k_{2z} = (\omega/c)\sqrt{\varepsilon_b}$ are the wave vectors of the first and second layers.

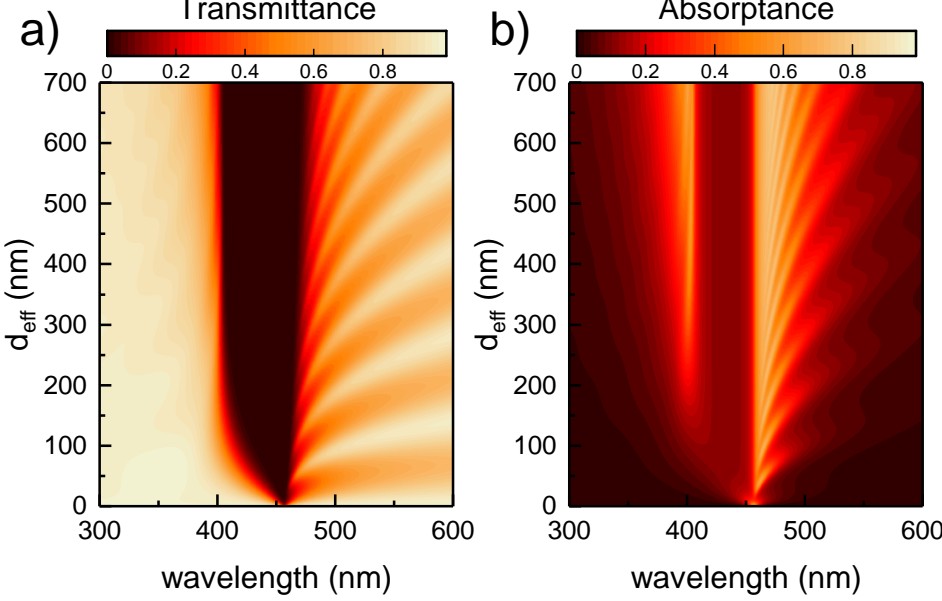

**Figure 4.** (**a**) Transmission and (**b**) absorption spectra of the NC film at different film thicknesses and incident radiation wavelengths. The filling factor is $f = 0.11$.

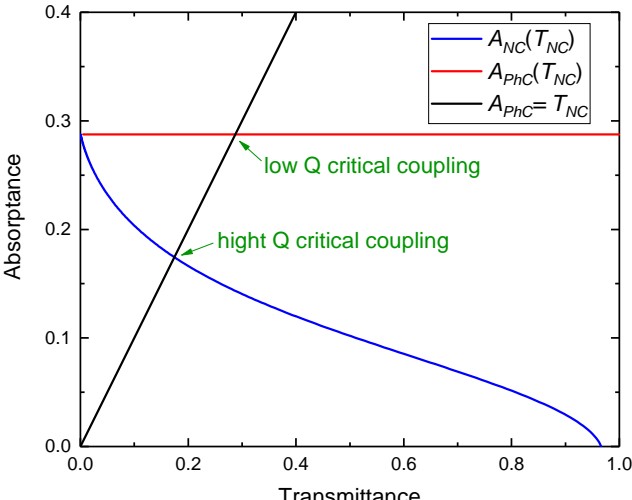

**Figure 5.** The dependence of absorptance of the NC film on its transmittance at different film thicknesses (blue line) and the dependence of the absorptance of an opaque NC film on the PhC transmittance at different numbers of PhC periods (red line). The point of intersection between the black and blue lines corresponds to the critical coupling condition (12), and the point of intersection between the black and red lines corresponds to Condition (13). The wavelength is $\lambda = 407.1$ nm.

It can be seen from the figures that in the scheme of TPP excitation through the NC, the critical coupling conditions are established at lower transmittances and absorptances, i.e., at lower energy relaxation rates, the sum of which determines the resonance spectral line width. Thus, the scheme of TPP excitation through the NC is more attractive, since under the critical coupling conditions, the resonance line and, consequently, absorption band are narrower. This comparison is contrary to the comparison of IR radiation TPP linewidths experimentally observed in [41]. This can be explained through the difference between metal dispersion laws for visible and IR radiations.

## 5. Numerical Calculation

To compare the two excitation schemes numerically, we calculate the transmittance spectra of the structures using the transfer matrix method [44]. The results of the calculation are presented in Figure 6.

It can be seen in Figure 6a that the critical coupling of the TPP with the incident field upon excitation through the NC is obtained at an NC layer thickness of $d_{eff} = 201$ nm, which is consistent with the above-described theory. In addition, the calculation showed that upon TPP excitation from the PhC side, the critical coupling is obtained at a number of PhC periods of $N = 3$. In this case, the spectral line for the first excitation scheme appears narrower, which is in good agreement with the data reported in the previous section.

The energy spectra of the structure for the two excitation schemes under the critical coupling conditions are shown in Figure 7.

Under the TPP excitation from the NC side, the maximum absorptance is observed at a wavelength of $\lambda = 407.1$ nm and under the excitation from the PhC side at a wavelength of $\lambda = 406.7$ nm. The effective permittivity at these wavelengths takes the values $\varepsilon_{eff} = 0.0094 + 0.0858i$ and $\varepsilon_{eff} = 0.0348 + 0.0843i$, respectively.

The spatial distributions of the local field intensity at the TPP wavelengths are presented in Figure 8.

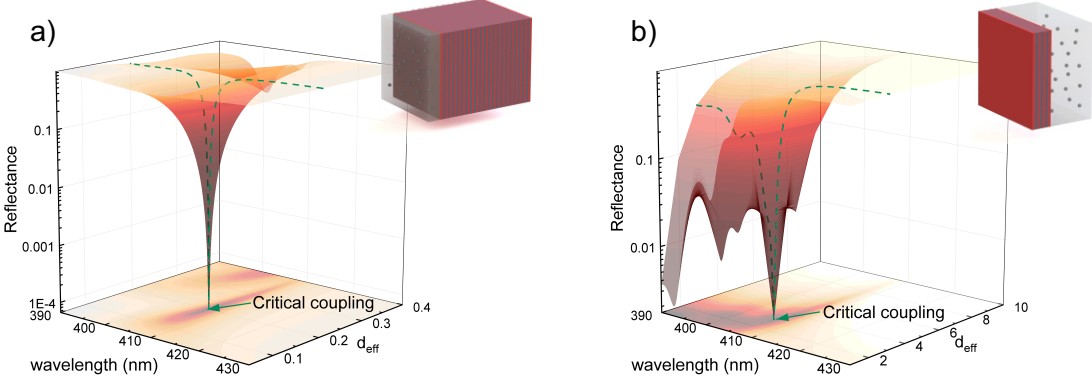

**Figure 6.** (**a**) NC-PhC and (**b**) PhC-NC reflectance spectra of the structure at different NC layer thicknesses $d_{eff}$ and a constant filling factor of $f = 0.11$. The green dashed line shows the reflectance spectra of the structures under critical coupling conditions.

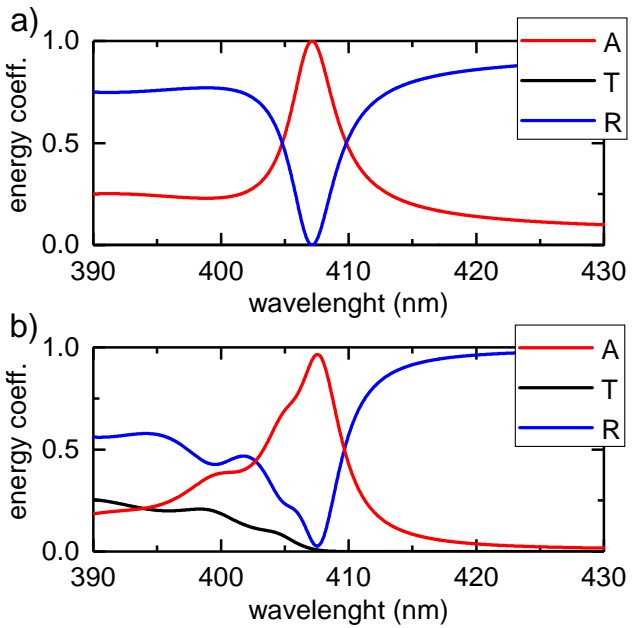

**Figure 7.** Transmittance (red line), reflectance (blue line), and absorptance (black line) spectra of the structure under the critical coupling conditions upon TPP excitation from (**a**) the NC and (**b**) PhC side. The NC layer thickness and number of PhC periods are $d_{eff} = 201$ nm, $N = 25$ and $d_{eff} = 700$ nm, $N = 3$, respectively.

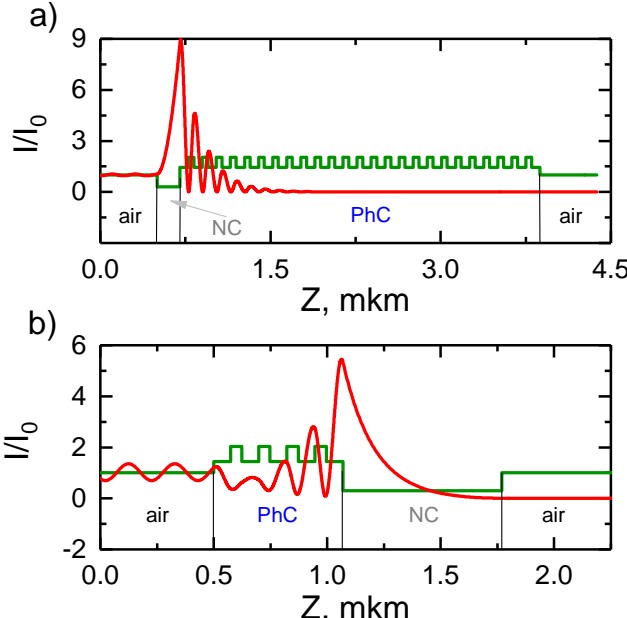

**Figure 8.** TPP in the region of positive $\varepsilon_{eff}$. Spatial distribution of the refractive index of the structure (green line) and local field intensity at the TPP wavelength (red line) for the TPP excitation from (**a**) the NC and (**b**) PhC side.

In both cases, the field is localized at the interface between the media in the region comparable with the wavelength. The localization value for the scheme of excitation through the NC is higher than that for the scheme of excitation through the PhC by a factor of 1.5.

It should be noted that under excitation from the PhC side, at the TPP wavelength (Figure 7b), 2.5% of the radiation falling onto the structure is not absorbed, but passes through the NC layer. This is related to the TPP formation in the range of small positive epsilon values, where the NC plays the role of a dissipative dielectric [45].

The zero transmission can be obtained by changing the TPP wavelength via changing the thickness of the PhC layer adjacent to the NC. As a result, the localized mode wavelength appears in the metal-like NC region, and the transmittance decreases (Figure 9).

The local field intensity at the TPP wavelength normalized to the input intensity is shown in Figure 9b. Here, the field localization is higher than in the case shown in Figure 8a by 20%. This confirms the fact that the critical coupling of the incident field with the TPP is obtained. Comparing with Figure 8b, we can see in Figure 9b that the skin layer of the metal-like NC is thinner, and about $10^{-6}$ of the incident radiation is transmitted at the TPP wavelength.

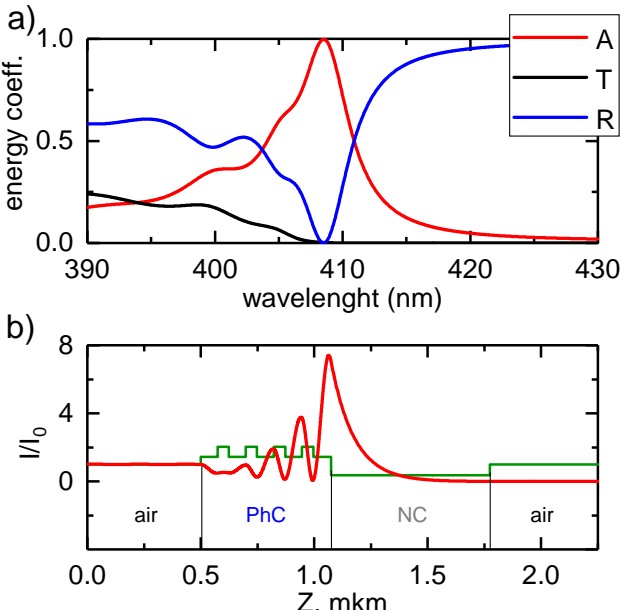

**Figure 9.** (**a**) TPP in the metal-like NC region $\varepsilon_{eff} < 0$. Transmittance (black line), reflectance (blue line), and absorptance (red line) spectra of the structure under the critical coupling conditions for the scheme of TPP excitation from the PhC side at a thickness of $d_{first} = 78$ nm of the layer adjacent to the PhC and (**b**) the spatial distribution of the local field intensity at the TPP wavelength.

## 6. Conclusions

We examined the optical properties of TPPs localized at the interface between a PhC and an NC with the near-zero permittivity. We studied two TPP excitation schemes: through a PhC and through an NC layer. Using the temporal coupled-mode theory, we determined the parameters of the structure corresponding to the critical coupling of the incident field with the TPP. It was established that under this condition, the entire radiation falling onto the structure is absorbed by it. It was demonstrated analytically and numerically that the scheme of TPP excitation through the NC is more attractive, since under the critical coupling conditions, the resonance line and, consequently, the absorption band is narrower. It was established that to obtain the 100% absorption of the incident radiation upon TPP excitation through the PhC, it is necessary to change the thickness of the first layer of PhC adjacent to the NC. This ensures the required phase shift of the wave and TPP localization in the region with the negative epsilon values. The proposed model can be used in designing narrowband absorbers based on Tamm plasmon polaritons localized in resonant PhC structures.

**Author Contributions:** R.G.B. performed the calculations, visualized the results, and drafted the manuscript. I.V.T. helped with the software and methods. S.Y.V. supervised the whole study and finalized the manuscript.

**Funding:** This research was funded by RFBR according to the research project No. 18-32-00053. I.V.T. and S.Y.V. acknowledge financial support from RFBR and MOST according to the research project No. 19-52-52006.

**Acknowledgments:** R.G.B. thanks Pavel Pankin for useful discussions about the couple mode theory.

**Conflicts of Interest:** The authors declare no conflict of interest.

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
