# Peer review of "Epsilon-Near-Zero Absorber by Tamm Plasmon Polariton"

_photonics, doi:10.3390/photonics6010028_

Round 1

Reviewer 1 Report

The manuscript contains interesting topics and proposes some theoretical predctions.

The combination of nanoparticle in tramparent matrix and photonics crystal is discussed to understand the criteria to exhibit Tamm plasmom. The topics is interesting and timely and may be published in this journal. However there are some points that are not clear and before considered to be included in to this journal it need to be considered. 

1) The title contains the term "epsilon near zero" but the necessity of NEZ is not clear to me. They deal with the permittivity feature shown as an inset of Fig 2. and this is feature is simply showing metallic feature. So it seems any meal is expected to show Tamm plasmon. Why ENZ is highlighted here? Can you explain the motivation in a clearer manner in the introduction and in discussions? 

2) In conclusion, it is stated that the thickness of the first PhC coupled to NC is necessary to be changed, to make 100% absorption. What does this mean? What does the first PhC mean? There is only one PhC in the two systems and there is no second PhC. Please spend more line to explain this.

Author Response

Responses to Reviewer Comments

Beginning from the following page, all of the amendments are marked in red if part of the revised manuscript is quoted in each of our responses. 

Referee 1.

The manuscript contains interesting topics and proposes some theoretical predctions.

The combination of nanoparticle in tramparent matrix and photonics crystal is discussed to understand the criteria to exhibit Tamm plasmom. The topics is interesting and timely and may be published in this journal. However, there are some points that are not clear and before considered to be included in to this journal it need to be considered.

Comment 1:

The title contains the term "epsilon near zero" but the necessity of NEZ is not clear to me. They deal with the permittivity feature shown as an inset of Fig 2. and this is feature is simply showing metallic feature. So it seems any meal is expected to show Tamm plasmon. Why ENZ is highlighted here? Can you explain the motivation in a clearer manner in the introduction and in discussions?

Response:

We thank the reviewer for this comment. A clarification of why this material is used in our paper was made in the introduction section

The patch is as follows:

It should be noted that the properties of ENZ materials are determined by their dispersion. For example, the real part of the dielectric permittivity may be less than the imaginary by two or more orders of magnitude. Moreover, the ENZ material can be received in visible, UV and IR spectral ranges [Maas, R.; Parsons, J.; Engheta, N.; Polman, A. Experimental realization of an epsilon-near-zero metamaterial at visible wavelengths. Nature Photonics2013, 7, 907–912. doi:10.1038/nphoton.2013.256; Chen, X.; Zhang, C.; Yang, F.; Liang, G.; Li, Q.; Guo, L.J. Plasmonic Lithography Utilizing Epsilon Near Zero Hyperbolic Metamaterial. ACS Nano 2017, 11, 9863–9868. doi:10.1021/acsnano.7b03584.21; Chen, C.; Wang, Z.; Wu, K.; Ye, H. Tunable near-infrared epsilon-near-zero and plasmonic properties of Ag-ITO co-sputtered composite films. Science and Technology of Advanced Materials2018, 19, 174–184. doi: 10.1080/14686996.2018.1432230.]. The classical plasmonic materials (silver, gold) do not have such dispersion properties. For this reason, ENZ materials are interesting for various plasmonics applications, for example, for tunable broadband absorbers [Anopchenko, A.; Tao, L.; Arndt, C.; Lee, H.W.H. Field-Effect Tunable and Broadband Epsilon-Near-Zero Perfect Absorbers with Deep Subwavelength Thickness. ACS Photonics 2018, 5, 2631–2637. doi:10.1021/acsphotonics.7b01373].

Comment 2:

In conclusion, it is stated that the thickness of the first PhC coupled to NC is necessary to be changed, to make 100% absorption. What does this mean? What does the first PhC mean? There is only one PhC in the two systems and there is no second PhC. Please spend more line to explain this.

Response:

This aspect has been clarified.

The sentence

… the thickness of the first PhC coupled to the NC.…

was replaced by

the thickness of the first layer of PhC adjacent to the NC.…

Reviewer 2 Report

In the manuscript "Epsilon-Near-Zero Absorber by Tamm Plasmon Polariton" the authors describe numerically and analytically how a Tamm-plasmon polariton can be established when the metal of the metal-photonic crystal-surface is not a pure material, but, a nano composite tuned to show an epsilon near zero. Then the authors tune different geometries to achieve full absorption of impinging light of a single frequency.

In general the research is well performed and the results are well presented. The involved theory is introduced nicely and I want to propose only minor changes: Between line 57 and 58 there are several additional lines, which could be fixed for the next version. Within these "secret" lines is the introduction of the filling factor f, where the "-" sign should be removed. Next, the Drude parameters in lines 60,61 should be moved directly after eq. (3). For eq. (4) I wish for a clear distinction between imaginary i and the sum index, which should be "l" - an index missing also at the first \tau.

I'm a little bothered by the amount of self-citation which is not helping in the understanding of the papers content. In principle the explanation that nano composites can behave like metals and therefore support Tamm plasmons has been made with citing [18]. References [19-22] are unnecessary and should be removed. The same seems true for Ref. [34]. On the other hand I propose to include a fundamental citation for the Maxwell-Garnett theory eq. (2).

Finally a comparison to other absorber concepts and prospects for real experiments would be a nice finish to the story. Therefore I propose minor revisions to be made before publication.

Author Response

Responses to Reviewer Comments

Beginning from the following page, all of the amendments are marked in red if part of the revised manuscript is quoted in each of our responses. 

Referee 2.

In the manuscript "Epsilon-Near-Zero Absorber by Tamm Plasmon Polariton" the authors describe numerically and analytically how a Tamm-plasmon polariton can be established when the metal of the metal-photonic crystal-surface is not a pure material, but, a nano composite tuned to show an epsilon near zero. Then the authors tune different geometries to achieve full absorption of impinging light of a single frequency.

In general the research is well performed and the results are well presented. The involved theory is introduced nicely and I want to propose only minor changes:

Comment 1:

Between line 57 and 58 there are several additional lines, which could be fixed for the next version. Within these "secret" lines is the introduction of the filling factor f, where the "-" sign should be removed. Next, the Drude parameters in lines 60,61 should be moved directly after eq. (3). For eq. (4) I wish for a clear distinction between imaginary i and the sum index, which should be "l" - an index missing also at the first \tau.

Response:

We thank the reviewer for the remark. This inaccuracy has been corrected

Comment 2: I'm a little bothered by the amount of self-citation which is not helping in the understanding of the papers content. In principle the explanation that nano composites can behave like metals and therefore support Tamm plasmons has been made with citing [18]. References [19-22] are unnecessary and should be removed. The same seems true for Ref. [34]. On the other hand I propose to include a fundamental citation for the Maxwell-Garnett theory eq. (2).

Response:

The references [19-22] and [34] has been removed. New citation was added. Corresponding renumbering of the following references was made.

Comment 3: Finally a comparison to other absorber concepts and prospects for real experiments would be a nice finish to the story. Therefore I propose minor revisions to be made before publication.

Response:

A comparison of presented TPP structures to other absorber types was made in introduction section.

The patch is as follows:

Moreover, tunable and broadband absorbers based on ENZ materials are realized in [Anopchenko, A.; Tao, L.; Arndt, C.; Lee, H.W.H. Field-Effect Tunable and Broadband Epsilon-Near-Zero Perfect Absorbers with Deep Subwavelength Thickness. AC  Photonics 2018, 5, 2631–2637. doi:10.1021/acsphotonics.7b01373; Yoon, J.; Zhou, M.; Badsha, M.A.; Kim, T.Y.; Jun, Y.C.; Hwangbo, C.K.  Broadband Epsilon-Near-Zero Perfect Absorption in the Near-Infrared. Scientific Reports 2015, 5. doi:10.1038/srep12788]. Also these materials can be used for TPP formation [Vetrov, S.Y.; Bikbaev, R.G.; Rudakova, N.V.; Chen, K.P.; Timofeev, I. Optical Tamm states at the interface between a photonic crystal and an epsilon-near-zero nanocomposite.Journal of Optics 2017, 19, 085103. doi:10.1088/2040-8986/aa75fb.]. However, to the best of our knowledge the possibility of realizing ENZ absorbers based on TPP has not been considered to date. This structure compared to metasurface absorbers [Landy, N.I.; Sajuyigbe, S.; Mock, J.J.; Smith, D.R.; Padilla, W.J.  Perfect Metamaterial Absorber. Physical Review Letters 2008, 100.  doi:10.1103/physrevlett.100.207402; Liu, N.; Mesch, M.; Weiss, T.; Hentschel, M.; Giessen, H. Infrared Perfect Absorber and Its Application As Plasmonic Sensor. Nano Letters 2010, 10, 2342–2348.  doi:10.1021/nl9041033.] has advantage of thickness, which allows for efficient removal of heat and protects the device from damage. Furthermore, TPP can be coupling with other types of localized modes [Pankin, P.S.; Vetrov, S.Y.; Timofeev, I.V.  Tunable hybrid Tamm-microcavity states. Journal of the Optical Society of America B 2017, 34, 2633. doi:10.1364/josab.34.002633; Brückner, R.; Sudzius, M.; Hintschich, S.I.; Fröb, H.; Lyssenko, V.G.; Leo, K. Hybrid optical Tamm states in a planar dielectric microcavity. Physical Review B 2011, 83, 033405.  doi:10.1103/PhysRevB.83.033405; Zhang, X.L.; Feng, J.; Han,  X.C.; Liu, Y.F.; Chen, Q.D.; Song,  J.F.;   Sun, H.B. Hybrid Tamm plasmon-polariton/microcavity modes for white top-emitting organic light-emitting devices. Optica 2015, 2, 579.  doi:10.1364/OPTICA.2.000579] which allows controlling its spectral properties and can be used for creation of tunable absorbers.

Reviewer 3 Report

The authors provide an interesting theoretical analysis on the design of absorbers at visible frequencies based on Tamm plasmon polaritons (TPPs) excited at the interface between a photonic crystal and a meta-dielectric nanocomposite, near the epsilon-near-zero (ENZ) point of the latter.

The authors give a very good review of the literature on TPPs, including their own past work. However, in my opinion they failed to relate their results to previous works on absorbers and ENZ media.

First, the design of absorbers it is a mature field. It has been known for decades how to design absorbers, narrowband or broadband, by matching the impedance of the device to that of free-space. See, for example:

[Munk, B. A., Munk, P., & Pryor, J. (2007). On designing Jaumann and circuit analog absorbers (CA absorbers) for oblique angle of incidence. IEEE Transactions on Antennas and Propagation, 55(1), 186-193.]

These techniqueshave been extended into the optical regime, including the visible frequencies:

[Li, Z., Butun, S., & Aydin, K. (2015). Large-area, lithography-free super absorbers and color filters at visible frequencies using ultrathin metallic films. Acs Photonics, 2(2), 183-188.]

[Landy, N. I., Sajuyigbe, S., Mock, J. J., Smith, D. R., & Padilla, W. J. (2008). Perfect metamaterial absorber. Physical review letters, 100(20), 207402.]

[Liu, N., Mesch, M., Weiss, T., Hentschel, M., & Giessen, H. (2010). Infrared perfect absorber and its application as plasmonic sensor. Nano letters, 10(7), 2342-2348.]

Since the device based on TPPs seems bulkier than any of these approaches, I urge the authors to review the state-of-the-art of absorbers and clarify whether if their approach brings anything new to the field.

Second, different phenomena have been associated with ENZ media. The present absorber operates near the ENZ frequency. However, it is not clear to me if it takes advantage of any of the characteristics of ENZ media, or if it could operate for any other frequency of the nanocomposite.

In summary, although I find interesting the analysis of TPPs for the implementation of absorber, I believe the authors should clarify what is the interest of this system within the context of absorbers and ENZ media.

Author Response

Responses to Reviewer Comments

Beginning from the following page, all of the amendments are marked in red if part of the revised manuscript is quoted in each of our responses. 

Referee 3.

The authors provide an interesting theoretical analysis on the design of absorbers at visible frequencies based on Tamm plasmon polaritons (TPPs) excited at the interface between a photonic crystal and a meta-dielectric nanocomposite, near the epsilon-near-zero (ENZ) point of the latter.

The authors give a very good review of the literature on TPPs, including their own past work. However, in my opinion they failed to relate their results to previous works on absorbers and ENZ media.

In summary, although I find interesting the analysis of TPPs for the implementation of absorber, I believe the authors should clarify what is the interest of this system within the context of absorbers and ENZ media.

Comment 1:

First, the design of absorbers it is a mature field. It has been known for decades how to design absorbers, narrowband or broadband, by matching the impedance of the device to that of free-space. See, for example:

·         [Munk, B. A., Munk, P., & Pryor, J. (2007). On designing Jaumann and circuit analog absorbers (CA absorbers) for oblique angle of incidence. IEEE Transactions on Antennas and Propagation, 55(1), 186-193.]

These techniqueshave been extended into the optical regime, including the visible frequencies:

·         [Li, Z., Butun, S., & Aydin, K. (2015). Large-area, lithography-free super absorbers and color filters at visible frequencies using ultrathin metallic films. Acs Photonics, 2(2), 183-188.]

·         [Landy, N. I., Sajuyigbe, S., Mock, J. J., Smith, D. R., & Padilla, W. J. (2008). Perfect metamaterial absorber. Physical review letters, 100(20), 207402.]

·         [Liu, N., Mesch, M., Weiss, T., Hentschel, M., & Giessen, H. (2010). Infrared perfect absorber and its application as plasmonic sensor. Nano letters, 10(7), 2342-2348.]

Since the device based on TPPs seems bulkier than any of these approaches, I urge the authors to review the state-of-the-art of absorbers and clarify whether if their approach brings anything new to the field.

Response:

We thank the reviewer for this comment.

We agree that TPP structure is bulkier than mentioned types of absorbers. But we think that this structure thickness is advantageous, allows for better removal of heat and protects the device from damage.

The patch is as follows:

Moreover, tunable and broadband absorbers based on ENZ materials are realized in [Anopchenko, A.; Tao, L.; Arndt, C.; Lee, H.W.H. Field-Effect Tunable and Broadband Epsilon-Near-Zero Perfect Absorbers with Deep Subwavelength Thickness. AC  Photonics 2018, 5, 2631–2637. doi:10.1021/acsphotonics.7b01373; Yoon, J.; Zhou, M.; Badsha, M.A.; Kim, T.Y.; Jun, Y.C.; Hwangbo, C.K.  Broadband Epsilon-Near-Zero Perfect Absorption in the Near-Infrared. Scientific Reports 2015, 5. doi:10.1038/srep12788]. Also these materials can be used for TPP formation [Vetrov, S.Y.; Bikbaev, R.G.; Rudakova, N.V.; Chen, K.P.; Timofeev, I. Optical Tamm states at the interface between a photonic crystal and an epsilon-near-zero nanocomposite.Journal of Optics 2017, 19, 085103. doi:10.1088/2040-8986/aa75fb.]. However, to the best of our knowledge the possibility of realizing ENZ absorbers based on TPP has not been considered to date. This structure compared to metasurface absorbers [Landy, N.I.; Sajuyigbe, S.; Mock, J.J.; Smith, D.R.; Padilla, W.J.  Perfect Metamaterial Absorber. Physical Review Letters 2008, 100.  doi:10.1103/physrevlett.100.207402; Liu, N.; Mesch, M.; Weiss, T.; Hentschel, M.; Giessen, H. Infrared Perfect Absorber and Its Application As Plasmonic Sensor. Nano Letters 2010, 10, 2342–2348.  doi:10.1021/nl9041033.] has advantage of thickness, which allows for efficient removal of heat and protects the device from damage. Furthermore, TPP can be coupling with other types of localized modes [Pankin, P.S.; Vetrov, S.Y.; Timofeev, I.V.  Tunable hybrid Tamm-microcavity states. Journal of the Optical Society of America B 2017, 34, 2633. doi:10.1364/josab.34.002633; Brückner, R.; Sudzius, M.; Hintschich, S.I.; Fröb, H.; Lyssenko, V.G.; Leo, K. Hybrid optical Tamm states in a planar dielectric microcavity. Physical Review B 2011, 83, 033405.  doi:10.1103/PhysRevB.83.033405; Zhang, X.L.; Feng, J.; Han,  X.C.; Liu, Y.F.; Chen, Q.D.; Song,  J.F.;   Sun, H.B. Hybrid Tamm plasmon-polariton/microcavity modes for white top-emitting organic light-emitting devices. Optica 2015, 2, 579.  doi:10.1364/OPTICA.2.000579] which allows controlling its spectral properties and can be used for creation of tunable absorbers.

Comment 2:

Second, different phenomena have been associated with ENZ media. The present absorber operates near the ENZ frequency. However, it is not clear to me if it takes advantage of any of the characteristics of ENZ media, or if it could operate for any other frequency of the nanocomposite.

Response:

We thank the reviewer for this comment. 

Certainly the absorber can operate for any other frequency of the nanocomposite where the real part of dielectric permittivity is negative. However, in this work we investigate the ENZ region of nanocomposite, because in this spectral range it has interesting dispersion properties. The plasmonic materials, such as gold and silver, don't have such features. Corresponding explanation was made in introduction section.

The patch is as follows:

It should be noted that the properties of ENZ materials are determined by their dispersion. For example, the real part of the dielectric permittivity may be less than the imaginary by two or more orders of magnitude. Moreover, the ENZ material can be received in visible, UV and IR spectral ranges [Maas, R.; Parsons, J.; Engheta, N.; Polman, A. Experimental realization of an epsilon-near-zero metamaterial at visible wavelengths. Nature Photonics2013, 7, 907–912. doi:10.1038/nphoton.2013.256; Chen, X.; Zhang, C.; Yang, F.; Liang, G.; Li, Q.; Guo, L.J. Plasmonic Lithography Utilizing Epsilon Near Zero Hyperbolic Metamaterial. ACS Nano 2017, 11, 9863–9868. doi:10.1021/acsnano.7b03584.21; Chen, C.; Wang, Z.; Wu, K.; Ye, H. Tunable near-infrared epsilon-near-zero and plasmonic properties of Ag-ITO co-sputtered composite films. Science and Technology of Advanced Materials2018, 19, 174–184. doi: 10.1080/14686996.2018.1432230.]. The classical plasmonic materials (silver, gold) do not have such dispersion properties.